# Effects of Dehydroepiandrosterone (DHEA) Supplementation on Ovarian Cumulus Cells following In Vitro Fertilization (IVF)/Intra-Cytoplasmic Sperm Injection (ICSI) Treatment—A Systematic Review

**DOI:** 10.3390/life13061237

**Published:** 2023-05-24

**Authors:** Woon Shu Yuan, Muhammad Azrai Abu, Mohd Faizal Ahmad, Marjanu Hikmah Elias, Abdul Kadir Abdul Karim

**Affiliations:** 1Advanced Reproductive Center, Department of Obstetrics and Gynaecology, UKM Medical Centre, Kuala Lumpur 56000, Malaysia; 2Faculty of Medicine & Health Sciences, Universiti Sains Islam Malaysia, Bandar Baru Nilai 71800, Malaysia

**Keywords:** cumulus cells, Dehydroepiandrosterone (DHEA), oocyte quality, ovarian function, in vitro fertilization

## Abstract

Despite many studies exploring the effects of DHEA supplementation, its application in IVF procedure continues to be a subject of debate owing to the inconsistent findings and the lack of rigorously designed, large-scale, randomized trials. Our review aims to explore the effectiveness of DHEA supplementation in ovarian cumulus cells following IVF/ICSI treatment. We conducted a literature search of Pub-Med, Ovid MEDLINE, and SCOPUS (inception to June 2022) for all relevant articles, including the keywords of “dehydroepiandrosterone/DHEA”, “oocyte”, and “cumulus cells”. From the preliminary search, 69 publications were identified, and following a thorough screening process, seven studies were ultimately incorporated into the final review. Four hundred twenty-four women were enrolled in these studies, with DHEA supplementation being administered exclusively to women exhibiting poor ovarian response/diminished ovarian reserve or belonging to an older age demographic. The intervention in the studies was DHEA 75–90 mg daily for at least 8–12 weeks. The only randomized controlled trial showed no difference in clinical or cumulus cell-related outcomes between the control and treatment groups. However, the remaining six studies (two cohorts, four case-controls) showed significant beneficial effects of DHEA in cumulus cell-related outcomes compared to the group (older age or POR/DOR) without DHEA supplementation. All studies revealed no significant difference in stimulation and pregnancy outcomes. Our review concludes that DHEA supplementation did show beneficial effect on ovarian cumulus cells in improving oocyte quality for women of advanced age or with poor ovarian responders.

## 1. Introduction

Globally, the prevalence of infertility is on the rise, impacting approximately 10% to 15% of couples within the reproductive age [1]. Assisted reproductive technology (ART) research has predominantly concentrated on female infertility for past decades, particularly in women of advanced maternal age. Infertility treatments involve the manipulation of gametes or embryos themselves, whereas in vitro fertilisation (IVF) treatments are the commonest treatment sought. The success of IVF procedures is intimately linked to the quantity of oocytes obtained, with oocyte quality being a critical determinant of favourable outcomes.

Androgens, crucial precursors for oestrogen synthesis in the ovary, hold a key physiological function in the progression and activity of preantral and antral follicles, encompassing the stimulation of granulosa cell (GC) proliferation. Evidence demonstrates that androgen receptor (AR) gene expression within primate ovaries is most pronounced in GCs of robust, expanding follicles, where its expression is enhanced by testosterone. The observed positive relationship between granulosa AR gene expression and cellular proliferation, coupled with the inverse correlation with apoptosis, suggests that androgens play a crucial role in early primate follicle development [2]. Hypoandrogenism, a potential etiological factor for reduced ovarian function in females, may precipitate premature ovarian failure, defined as the cessation of ovarian function in women under the age of 40 [3,4].

Over the years, various methodologies have been put forth to optimize ovarian function and receptiveness to stimulation, amplify fertilization outcomes, refine embryo quality, and elevate the probability of implantation and successful pregnancy. One of the proposed strategies is to provide supplements, such as dehydroepiandrosterone (DHEA), before ovarian stimulation to enhance the quality of oocytes. The adrenal prohormone DHEA and its sulphate conjugate, 3β-hydroxy-5-androsten-17-one, represent C19 endogenous steroids predominantly synthesized by the adrenal zona reticularis (50%) and ovarian theca cells (20%); with an additional 30% derived from circulating DHEAs [5]. DHEA and testosterone are postulated to enhance conception probabilities by exerting a positive influence on follicular responsiveness to gonadotropin stimulation. This results in elevated oocyte yields, subsequently increasing the likelihood of a successful pregnancy [6]. Casson et al. initially hypothesized that the exogenous administration of DHEA could potentially rejuvenate ovarian follicular sex steroidogenesis in women of advanced age. Oral administration of physiological DHEA led to a considerable elevation in serum insulin-like growth factor-I concentrations, a contributing factor in the mediation of androgen-induced follicular growth [7]. Multiple researchers have documented the advantageous effects of DHEA supplementation prior to IVF cycles in women with poor ovarian response (POR) who underwent IVF/Intracytoplasmic sperm injection (ICSI) procedures. For women with POR undergoing ART, pre-treatment utilizing DHEA or testosterone may correlate with enhanced live birth rates, supported by a moderate overall quality of evidence [6].

For the past decades, the research scope has been extended to changes in the whole ovarian microenvironment in female infertility. Numerous studies have examined the association between morphological attributes of follicles or cumulus-oocyte complexes (COCs) and the developmental competence of oocytes.

Cumulus cells (CCs), the somatic cells that envelop oocytes to create COC, serve vital functions in follicular development, including oocyte maturation, meiotic resumption, and ovulation.

The morphological state of the cumulus mass in terms of its expansion has been used to determine the maturity of COCs. CCs represent a promising resource for dependable markers in predicting oocyte quality, and the assessment of CC transcript levels offers valuable insights that complement morphological selection [8]. Evaluating CCs constitutes an effective method to obtain additional information on the morphological and metabolic evaluation of oocytes to appropriately select those with a high chance of fertilisation and development [8].

Numerous studies have been conducted, substantiating the concept that the expressions of potential marker genes in CCs are correlated with (i) oocyte competence, (ii) embryo quality, (iii) pregnancy outcome, and (iv) live birth. The multivariable models grounded in CC gene expression hold the potential to predict embryo progression and subsequent pregnancy outcomes [9].

Recent research has indicated that DHEA supplementation could potentially impact gene expressions in CCs, which are implicated in extracellular matrix (ECM) formation, cellular development, differentiation and apoptosis regulation among women with POR. Additionally, the method employed for CCs isolation is regarded as non-invasive due to its execution during oocyte retrieval, in which it does not cause any disruptions to the ongoing IVF cycle. Thus, examining the impact of DHEA on the gene expression of CCs could serve as an effective approach to pinpoint the precise mechanisms of DHEA within the follicular microenvironment.

Although DHEA is extensively employed as a supplementary component in IVF treatment protocols for women with diminished ovarian reserve (DOR) and POR globally, its efficacy remains a subject of debate. This is primarily due to the absence of robust evidence to support its use, and the precise reproductive mechanism of DHEA continues to be conjectural [10]. In this review, our objective is to investigate the impacts of DHEA supplementation on CC-associated functions among women undergoing IVF/ICSI procedures.

## 2. Methods

### 2.1. Search Strategy

We conducted a comprehensive literature search utilizing the following electronic databases from inception to 30 June 2022: PUBMED, OVID MEDLINE, and SCOPUS. The Medical Subject Heading (MeSH) terms such as “Dehydroepiandrosterone”, “Oocyte Quality”, and “Cumulus Cell” were used as the keywords in all fields. Synonyms for keywords were derived utilizing MeSH terms from the National Library of Medicine (NLM). The search strategy incorporated a combination (using “AND”) of the following sets of keywords: (i) Dehydroepiandrosterone” OR “Dehydroisoandrosterone” OR “5-Androsten-3-beta-hydroxy-17-one” OR “5-Androsten-3-ol-17-one” OR “Androstenolone”, (ii) “Oocyte Quality” OR “Ovocyte”, (iii) “Cumulus Cell” OR “Cumulus Granulosa Cells” OR “Granulosa Cells, Cumulus”.

### 2.2. Inclusion Criteria

We included all randomized controlled trials (RCTs), case-control, and cohort studies into our analysis. Studies deemed incomplete or RCTs exhibiting evidence of inadequate sequence generation were excluded. We included women undergoing IVF or ICSI treatment. Eligibility for inclusion was limited to studies employing DHEA as an adjunct treatment in comparison to a placebo or no intervention.

### 2.3. Exclusion Criteria

Women who had previously undergone oophorectomy, pelvic irradiation/cytotoxic exposure for malignancy, or that were on hormonal treatment were excluded from this review. We also excluded women who are using donor oocytes.

### 2.4. Selection of Studies

Following the primary search, all duplicated studies were removed using EndNote version 20.0.1. Two review authors (W.S.Y. and M.A.A.) independently assessed the titles and abstracts of studies identified through the search process for potential relevance, adhering to the predefined inclusion and exclusion criteria and intervention of the studies. Any studies that were not applicable were excluded. The selected articles were subjected to full-text screening.

### 2.5. Data Extraction and Management

For this review, we created a Microsoft Excel sheet specifically designed for data screening, assessment, and extraction. This Excel sheet encompassed details of all pertinent trial characteristics. The extracted data comprised demographic information (study type, trial period, number of participants), inclusion and exclusion criteria, interventions (treatment protocol type, dosage, and duration of treatment of intervention), and outcome data (clinical outcomes, cumulus-cell related outcomes). In instances of disagreements concerning selection during screening or eligibility by consensus, or when consensus could not be reached, the matter was referred to a third review author for resolution (M.F.A.).

### 2.6. Assessment of Risk of Bias

We independently evaluated the risk of bias in the included studies by employing the JBI Critical Appraisal assessment tool [11]. Each of the included studies was evaluated utilizing the specific assessment tool tailored to the respective study design. Two review authors conducted the risk of bias assessment (W.S.Y. and M.A.A.). All studies in our review obtained a minimum fair score (60–75%). The results of risk of bias assessment are stated in Table 1, Table 2 and Table 3 Any disagreements regarding the bias risk assessment were referred to the third review author (M.F.A.).

### 2.7. Protocol Registration

Our manuscript was formulated based on the Preferred Reporting Items for Systematic Reviews (PRISMA) recommendation (Figure 1). Our study was registered under the Prospective International Register of Systematic Reviews (PROSPERO) (Reg. No. CRD42022354272).

## 3. Results

### 3.1. Result of Search

The searches retrieved 69 articles. After removing 19 duplicated studies, 50 articles were screened for the title and abstracts. Fourteen articles were animal-related studies, and 26 were unrelated to DHEA/oocyte quality. Out of eight potentially eligible studies, seven were ultimately selected for the final review after a thorough examination of the full-text articles. The details of the selection process are illustrated in the Figure 1.

### 3.2. Characteristics of Included Studies

Among the seven included studies, one was a double-blinded randomised controlled trial (RCT), and the remaining six compromised prospective cohort/case-control studies. Study durations ranged from 8 months to 2 years. Two studies included only POR women who met the Bologna criteria [12], and the RCT recruited patients who were predicted to have DOR. The remaining four studies included all women who underwent IVF within the study period in their respective centres; the women were further categorised based on their age (young versus advanced age) or ovarian response (normal ovarian response (NOR) versus POR).

In the seven studies included, 424 women were recruited, and 143 of them received DHEA supplementation. All included studies involved intervention with DHEA (75–90 mg/day in one or divided dose) for at least 8–12 weeks before ovarian stimulation for the next IVF cycle, except for one small study where no details on the intervention were stated. Most studies excluded women with a history of oophorectomy, history of exposure to cytotoxic treatment or pelvic irradiation for malignancy, as well as those taking herbal medications or other hormonal agents. All women in the same study had a similar treatment protocol (including ovarian stimulation, trigger agent used, oocyte pick-up timing, embryo quality assessment, luteal phase support/post embryo transfer (ET) treatment, post-ET follow-up, CC collection, and analysis). (Table 4)

### 3.3. CC-Related Outcomes

Our review divided the results of the included studies into CC-related (Table 5 and clinical outcomes (Table 6). All seven studies investigated the effects of DHEA on CCs by exploring different aspects, including (i) expressions of different types of genes involved in ECM formation, follicular growth and maturation, regulation of cell differentiation and apoptosis; (ii) mitochondrial function; (iii) molecular metabolic mechanism. Table 5 shows the molecular mechanisms affected by DHEA supplementation.

Tsui et al. (2014) conducted a prospective study over a 10-month period that involved 10 women who met the Bologna criteria [12] for POR; they investigated alterations in various genes before and after DHEA supplementation with the intent to offer molecular substantiation for the benefits associated with DHEA supplementation [13]. In the initial study design, a total of 24 genes associated with ECM formation, cellular development, differentiation, apoptosis and other undetermined functions within CCs were examined. Out of the 24 CC genes, 18 demonstrated significant alterations, although 9 could not be validated. Another six genes did not exhibit significant differences pre- and post-DHEA supplementation. The remaining nine genes and their corresponding primers were further explored in the study, and all of them displayed a statistically significant difference following DHEA supplementation (all *p* < 0.05). Genes involved in ECM formation (hyaluronic acid synthase 2 (*HAS2*)*, Versican* (*VCAN*), and (*THBS1*)) were upregulated in a group with DHEA supplementation, and genes related to cellular development and differentiation upregulation (*RUNX2*, *CBX3*, and *TRIM28*) were all downregulated. Likewise, the expressions of *BCL-2* and *BAX,* which are involved in apoptosis regulation, decreased post-DHEA supplementation.

The RCT conducted by Narkwichean et al. (2017) also investigated several target genes involved in oocyte quality determination. They evaluated the possible mechanism of DHEA action by determining the mRNA expression level of 10 markers in CCs and GCs [14]. The findings indicated no discernible disparities in mRNA expression levels of target genes in CCs and GCs derived from ovulatory follicles when comparing the treatment and control groups. For patients who had ICSI, no significant changes were detected in the expression of eight cumulus mRNA transcripts (*Ar*, *prostaglandin synthase 2* (*PTGS2*), *HAS2*, *PTX3*, *Gremlin 1* (*GREM1*), *AREG*, *EREG,* and *BTC*) in the DHEA group, compared with the control group (*p* > 0.05 by Mann–Whitney U test). Similarly, the gene expressions within GCs exhibited no significant differences between the treatment and control groups.

Another small prospective cohort study by Lin et al. (2017) involved six POR women as defined by Bologna Criteria [12] and investigated the potential protective effects of DHEA on CCs through the reduction of apoptosis and enhancement of mitochondrial function [15]. Significantly lower expressions of pro-apoptotic molecules (cytochrome c, caspase-3, and caspase-9) were observed after DHEA supplementation (*p* < 0.05). In addition, mitochondrial dehydrogenase activity significantly increased (*p* < 0.0001). The authors additionally noted a reduced expression of *BAX* and an elevated expression of *BCL-2* subsequent to DHEA supplementation although no statistical significance was detected. These results were further supported by one of the included case-control study conducted by Lin et al. (2017), which involved 131 women, including 59 women with NORs and 72 women with PORs [16]. The authors found that the mRNA levels of *BAX*, *BAD*, caspase-3, caspase-9, and cytochrome c were all significantly reduced in the CCs obtained from the POR women who received DHEA supplementation compared to those from POR women without DHEA supplementation. A significantly lower proportion of apoptotic cells was also observed in comparison to the POR group (9.7% vs. 85.7%; *p* < 0.001). The same study further demonstrated that DHEA supplementation significantly elevated the expression of TFAM gene, which in turn enhanced mitochondrial dehydrogenase activity and mitochondrial mass in CCs for PORs. Hou et al. verified the association between *PGAM5* and other mitochondrial dynamic proteins (which may affect the aging process of germ cells) in their study, and the results displayed a significantly lowered level of *PGAM5* expression after DHEA administration [17].

A case-control study incorporated in this review examined the clinical advantages of DHEA in older patients as well as its anti-senescence impact on CCs [18]. In this study, the older group receiving DHEA supplementations exhibited a significantly lower proportion of SA-β-gal-positive cells compared to the older groups without DHEA supplementation (47.4% vs. 67.5%, *p* < 0.0001).

Li et al. (2021) explored the biological importance of the impact of DHEA on CCs by screening 306 genes with significant changes and uploading them to Metascape software (Metascape Ltd., London, UK) for functional enrichment analysis [19]. A meta-landscape analysis was conducted to identify higher-scoring markers among the differentially expressed genes with protein–protein interaction (PPI) enrichment. The authors noted that PPI clusters encompassed numerous signalling pathways and enhanced the interaction related to various glucose metabolism pathways, tricarboxylic acid (TCA) cycle, apoptosis, and monocarboxylic acid metabolism. The authors proposed that DHEA supplementation might rejuvenate oocytes by stimulating energy metabolism within CCs and GCs, subsequently transferring energy to the oocytes. The study revealed notable disparities between cells regarding the quantity of metabolites derived from glycolysis and the TCA cycle. Pyruvate, which regulated the key gene of acetyl-CoA and PDHA, was significantly increased in the DHEA group compared to the aged group. These findings imply that DHEA prompts alterations in glucose metabolism and the TCA cycle in aging CCs. Real-time assessment of the oxygen consumption rate demonstrated that cells from patients receiving DHEA restored normal cellular respiration, including maximum oxygen consumption rate (*p* < 0.01), ATP conversion rate (*p* < 0.05), and spare capacity (*p* < 0.05). These observations indicate that DHEA administration could potentially regulate cellular mitochondria, thereby enhancing the mitochondrial turnover rate and increasing intracellular energy levels.

**Table 1 life-13-01237-t001:** JBI critical appraisal of the included randomized controlled trials.

	Study	Narkwichean et al., 2017 [14]
Q1	Was true randomization used for assignment of participants to treatment groups?	Yes
Q2	Was allocation to treatment groups concealed?	Yes
Q3	Were treatment groups similar at the baseline?	Yes
Q4	Were participants blind to treatment assignment?	Yes
Q5	Were those delivering treatment blind to treatment assignment?	Yes
Q6	Were outcomes assessors blind to treatment assignment?	Yes
Q7	Were treatment groups treated identically other than the intervention of interest?	Yes
Q8	Was follow up complete and if not, were difference between groups in terms of their follow up adequately described and analyzed?	Yes
Q9	Were participants analyzed in the groups to which they were randomized?	Yes
Q10	Were outcomes measures in the same way for treatment groups?	Yes
Q11	Were outcomes measured in a reliable way?	Yes
Q12	Was appropriate statistical analysis used?	Yes
Q13	Was the trial design appropriate, and any deviations from the standard RCT design accounted for in the conduct and analysis of the trial?	Yes
	Total Score	100%

**Table 2 life-13-01237-t002:** JBI critical appraisal of the included cohort studies.

	Study	Tsui et al., 2014 [13]	Lin et al., 2017 [15]
Q1	Were the two groups similar and recruited from the same population?	Yes	Yes
Q2	Were the exposures measured similarly to assign people to both exposed and unexposed groups?	Yes	Yes
Q3	Was the exposure measured in a valid and reliable way?	Yes	Yes
Q4	Were confounding factors identified?	No	Yes
Q5	Were strategies to deal with confounding factors stated?	No	Yes
Q6	Were the groups/participants free of the outcome at the start of the study?	Not applicable	Not applicable
Q7	Were the outcomes measured in a valid and reliable way?	Yes	Yes
Q8	Was the follow up time reported and sufficient to be long enough for outcomes to occur?	Yes	Yes
Q9	Was follow up complete, and if not, were the reasons to loss to follow up described and explored?	Yes	Yes
Q10	Were strategies to address incomplete follow up utilized?	Not applicable	Not applicable
Q11	Was appropriate statistical analysis used?	Yes	Yes
	Total Score	64%	82%

**Table 3 life-13-01237-t003:** JBI critical appraisal of the included case control studies.

	Study	Lin et al., 2017 [18]	Lin et al., 2017 [16]	Li et al., 2021 [19]	Hou et al., 2022 [17]
Q1	Were the groups comparable other than the presence of disease in cases or the absence of disease in controls?	Yes	Yes	Yes	Yes
Q2	Were cases and controls matched appropriately?	Yes	Yes	Yes	Yes
Q3	Were the same criteria used for identification of cases and controls?	Yes	Yes	Yes	Yes
Q4	Was exposure measured in a standard, valid and reliable way?	Yes	Yes	Yes	Yes
Q5	Was exposure measured in the same way for cases and controls?	Yes	Yes	Yes	Yes
Q6	Were confounding factors identified?	Unsure	Unsure	Unsure	Unsure
Q7	Were strategies to deal with confounding factors stated?	Unsure	Unsure	Unsure	Unsure
Q8	Were outcomes assessed in a standard, valid and reliable way for cases and controls?	Yes	Yes	Yes	Yes
Q9	Was the exposure period of interest long enough to be meaningful?	Yes	Yes	Yes	Yes
Q10	Was appropriate statistical analysis used?	Yes	Yes	Yes	Yes
	Total Score	80%	80%	80%	80%

**Table 4 life-13-01237-t004:** Characteristics of the included studies.

No	Author, Year	Study Design	Participants	Inclusion Criteria	Exclusion Criteria	Intervention	Duration of Intervention
1	Narkwichean et al., 2017 [14]	Randomized Controlled Trial	Total: 52 (27 DHEA, 25 placebo)	- Aged > 23 years - Predicted Diminished Ovarian Reserve (DOR) by AFC < 10 and/or serum AMH < 5 pmol/L - Regular menstrual cycle	- Obesity with BMI > 35 kg/m^2^ - History of oophorectomy - With untreated hydrosalpinx, endometrial polyp or submucous myoma at beginning of treatment - With endocrinological disorders for, e.g., thyroid/adrenal disease - Allergy to DHEA - Treated with insulin for diabetic management	DHEA 75 mg/day	Minimum 12 weeks
2	Tsui et al., 2014 [13]	Prospective Cohort	Total: 10	PORs, met Bologna Criteria [12]	- History of ovarian cystectomy - History of oophorectomy - History of cytotoxic treatment or pelvic irradiation for malignancy - Taking herbal drugs or other hormonal agents	DHEA(exact dose not mentioned)	NA
3	Lin et al., 2017 [15]	Prospective Cohort	Total: 6	PORs, met Bologna Criteria [12] and had failed IVF cycle	- History of oophorectomy - History of cytotoxic treatment or pelvic irradiation for malignancy - Taking herbal drugs or other hormonal agents	DHEA 90 mg/day	Minimum 2 months
4	Lin et al., 2017 [18]	Prospective Case-Control	Total: 88≤37 yrs: 30≥37 yrs: 58 (28 with DHEA, 30 without)	Infertile women underwent IVF within study period	- History of oophorectomy - History of exposure to cytotoxic or pelvic irradiation for malignancy - Taking herbal drugs or other hormonal agents	DHEA 90 mg/day	Minimum 2 months
5	Lin et al., 2017 [16]	Prospective Case-Control	Total:131NORs: 59PORs: 72 (34 with DHEA, 38 No DHEA)	- Normal Ovarian Response (NORs) - PORs, met Bologna Criteria [12]	Not mentioned	DHEA 90 mg/day	8–16 weeks
6	Li et al., 2021 [19]	Prospective Case-Control	Total: 77≤ 38 yrs: 32>38 yrs: 45(20 with DHEA, 25 without)	- Infertile patient underwent IVF within study period	- History of oophorectomy - Donor cycle - Had pelvic radiotherapy or chemotherapy - Taken hormonal therapy within the last 3 months	DHEA 25 mg/^3^ times daily	Minimum 8 weeks
7	Hou et al., 2022 [17]	Prospective Case-Control	Total:60NORs: 22PORs: 38 (18 with DHEA, 20 No DHEA)	- NORs - PORs, met Bologna Criteria [12]	- History of oophorectomy - Donor cycle - Had pelvic radiotherapy or chemotherapy - Taken hormonal therapy within the last 3 months	DHEA 25 mg/^3^ times daily	Minimum 2 months

**Table 5 life-13-01237-t005:** Molecular mechanism affected by treatment with DHEA.

No	Author	Gene Tested	Outcomes			Pathway Involved
			Upregulated	Downregulated	No Difference	
1	Narkwichean et al. [14]	(i). PTX 3 (ii). HAS2 (iii). PTGS2 (iv). GREM1 (v). AREG (vi). ERED (vii). BTC (viii). EGF-like signalling molecules	-	-	*PTGS2*, *HAS2*, *PTX3*, *GREM1*, *AREG*, *EREG*, *BTC*	- Molecular markers of oocyte quality
2	Tsui et al. [13]	(i). HAS2 (ii). VCAN (iii). THBS2 (iv). RUNX2 (v). CBX3 (vi). TRIM28 (vii). BCL (viii). BAX (ix). ANKRD57	*HAS2*, *VCAN*, *THBS1*	*RUNX2*, *CBX3*, *TRIM28*, *BCL2*, *BAX*, *ANKRD57*	-	- Extracelluar Matrix formation - Cell development and differentiation - Apoptosis regulation
3	Lin et al. [15]	(i). BCL2 (ii). BAX (iii). Cytochrome c (iv). Caspase 3 (v). Caspase 9 (vi). Mitochondrial dehydrogenase activity	*BCL2*,Mitochondrial dehydrogenase activity	*BAX*, Cytochrome *c*, Caspase *3*, Caspase *9*	-	- Apoptosis regulation - Mitochondrial activity
4	Lin et al. [16]	(i). BAX (ii). BAD (iii). BCL2 (iv). Cytochrome c (v). Caspase 3 (vi). Caspase 9 (vii). TFAM	*BCL2* *TFAM*	*BAX*, *BAD*, Cytochrome *c*, Caspase *3*, Caspase *9*	-	- Apoptosis regulation - Mitochondrial activity
5	Hou et al. [17]	*PGAM5*		*PGAM5*	-	- Mitochondrial activity → Mitochondrial fission

**Table 6 life-13-01237-t006:** Clinical outcome of the included studies.

Study Author	Narkwichean et al. [14]	Tsui et al. [13]	Lin et al. [15]	Lin et al. [18]	Lin et al. [16]	Li et al. [19]	Hou et al. [17]
**Comparison**	DHEA vs. Non-DHEA	Post-DHEA	Post vs. Pre-DHEA	Older group DHEA vs. No DHEA	PORs group DHEA vs. No DHEA	Older group DHEA vs. No DHEA	PORs group DHEA vs. No DHEA
**Stimulation Outcomes**							
Stimulation Duration (Days)	12.5 (10–17) vs. 13 (10–14); *p* = 0.81	NA	10.3 ± 2.2 vs. 9.8 ± 2.5; *p* = 0.64	10.9 ± 1.9 vs. 10.3 ± 2.2	NA	NA	10.5 ± 1.5 vs. 12.1 ± 2.2
Gonadotrophin doses (IU)	3801.6 ± 1007.9 vs. 3802.2 ± 678.9; *p* = 0.99	NA	3150.0 ± 264.02 vs. 2950.0 ± 745.0; *p* = 0.49	3139.3 ± 595.7 vs. 2910.0 ± 813.3	NA	NA	2795.4 ± 684.1 vs. 2775.1 ± 806.9
**Cycle Outcomes**							
N. of oocytes retrieved	med (IQR): 4, 0–18 vs. 4, 0–15; *p* = 0.54	*p* < 0.01	3.17 ± 1.60 vs. 2.00 ± 1.10; *p* = 0.17	3.5 ± 2.1 vs. 2.4 ± 1.3	3.5 ± 2 vs. 2.3 ± 1.2	5.2 ± 1.4 vs. 3.2 ± 2.1; *p* < 0.05	3.8 ± 2.1 vs. 3.2 ± 2.2; *p* < 0.05
N. of MII oocytes retrieved	NA	NA	1.67 ± 0.82 vs. 0.50 ± 0.55; *p* < 0.05	2.3 ± 1.7 vs. 1.2 ± 1.0	2.2 ± 1.6 vs. 1.1 ± 0.9	2.3 ± 1.5 vs. 1.8 ± 1.7	2.7 ± 1.2 vs. 1.4 ± 2.5
Fertilization rate (%)	64.5 ± 24.9 vs. 48.0 ± 30.4; *p* = 0.052	NA	75.6 ± 28.5 vs. 22.2 ± 27.2; *p* < 0.01	76.8 vs. 55.4; *p* < 0.05	75.9 vs. 58.8; *p* < 0.05	71.7 ± 22.0 vs. 67.8 ± 21.2	76.5 ± 21.2 vs. 66.2 ± 11.2
N. of day 3 embryos	NA	*p* < 0.0001	2.17 ± 0.98 vs. 0.67 ± 0.82; *p* < 0.05	NA	NA	3.4 ± 1.6 vs. 1.7 ± 2.1; *p* < 0.05	3.5 ± 1.5 vs. 1.5 ± 2.2; *p* < 0.05
N. of top-quality D3 embryos	NA	NA	NA	1.2 ± 1.1 vs. 0.3 ± 0.6	1.2 ± 1.1 vs. 0.3 ± 0.6; *p* < 0.05	2.4 ± 1.7 vs. 0.7 ± 1.2; *p* < 0.05	1.9 ± 1.7 vs. 0.7 ± 1.5; *p* < 0.05
**Pregnancy Outcomes**							
Clinical pregnancy rate (%)	8 ± 30 vs. 9 ± 36; *p* = 0.63	NA	NA	17.7 vs. 4.8	18.7 vs. 5.2	26.3 vs. 16	27.7 vs. 15.0
Ongoing pregnancy rate (%)	NA	NA	NA	NA	15.6 vs. 2.6	26.3 vs. 16	22.2 vs. 15.0
Live birth rate (%)	7 ± 26 vs. 8 ± 32; *p* = 0.63	NA	NA	NA	12.9 vs. 2.6	16.7 vs. 12	22.2 vs. 10.0

NA: not availaible.

### 3.4. Clinical Outcomes

We subcategorised clinical outcomes into stimulation, cycle, and pregnancy outcomes (Table 3). The sole RCT in this review showed no statistically significant differences in all three clinical outcome subcategories in groups with and without DHEA supplementation [14]. Similarly, the other six studies showed no differences in the stimulation outcomes. The duration of ovarian stimulation and the total dose of gonadotropin used for the cycle showed no significant differences among the groups involved in the respective studies.

For the cycle outcomes, three studies observed a statistically significant increased number of oocytes retrieved [13,17,19], fertilisation rate [15,16,18], and the number of day-3 embryos [13,15,17] in groups treated with DHEA supplementation. Three of the recent case-control studies also showed a significant increase in top-quality day-3 embryos in the aged [19] and POR groups [16,17] after DHEA supplementations. Five of the seven studies presented pregnancy outcome results. Apart from the RCT, four other studies observed increased clinical pregnancy, ongoing pregnancy, and live birth rates among the older age group or women with poor ovarian response who were administered DHEA supplementation in comparison to those who did not receive DHEA supplementation. However, these results did not achieve statistical significance.

## 4. Discussion

Several studies have highlighted the positive impact of DHEA supplementation on enhancing embryo quality and the likelihood of pregnancy in women with diminished ovarian reserve (DOR) or poor ovarian response (POR). These beneficial effects are thought to be due to the potential role of DHEA in modulating the ovarian environment, including the regulation of gene expression in cumulus cells and the promotion of energy metabolism in both cumulus cells and granulosa cells. The use of DHEA in ovarian stimulation was first reported by Barad and Gleicher in 2005. Their study demonstrated that DHEA supplementation improved ovarian stimulation outcomes in a woman of advanced reproductive age who underwent multiple stimulation cycles for embryo cryopreservation and aneuploidy screening [20]. They proposed that DHEA, acting as a precursor to androgens, might potentially elevate the concentrations of androgens within the follicles. [20]. One prospective cohort study conducted in Beijing Hospital by Hu and colleagues involved 103 women with DOR; it showed that supplementation with DHEA has the potential to enhance the expression of androgen receptors in preovulatory granulosa cells both in vivo and in vitro settings [21]. The authors proposed that the selective positive impact of DHEA on ovarian response in women with DOR might be attributed to the upregulation of androgen receptor and follicle-stimulating hormone receptor expression in granulosa cells. 

One of the critical steps for oocyte maturation depends on the connection between oocytes and CCs. Therefore, changes in the gene expression of cumulus cells in women with poor ovarian response following DHEA supplementation could potentially promote oocyte maturation. McKenzie et al. (2004) showed that the expression levels of *PTGS2*, *HAS2*, and *GREM1* genes are associated with morphological and physiological features, offering an innovative method for predicting human embryo [22]. This finding was further supported by Gebhardt et al. (2011), who demonstrated that the gene expression profile of metabolic (*PFKP*), signalling (*PTGS2* and *GREM1*) and extracellular matrix (*VCAN*) components in cumulus cell masses could potentially serve as indicators for oocytes with elevated developmental potential. This, in turn, may result in improved implantation rates and increased developmental capacity throughout pregnancy [23]. In a related study, Shen et al. (2020) concluded that the expression level of VCAN in cumulus cells exhibited a positive correlation with early embryo morphology scores. Furthermore, this parameter could be utilized to evaluate oocyte developmental competence, supplementing traditional embryonic morphological assessments [24]. The cohort study by Tsui et al. (2014) included in this review showed that nine genes in CCs of recruited women significantly differed after DHEA supplementation. Three of the genes (*HAS2*, *VCAN*, and *THBS1*), which are involved in ECM formation, were upregulated. Inversely, three genes (*RUNX2*, *CBX 3*, and *TRIM 28*) related to cell development and differentiation upregulation were downregulated [13].

Apoptosis, a biologically significant and genetically regulated form of cell death, plays a crucial role in ovarian function and development. This mechanism involves the interaction between pro-apoptotic and pro-survival molecules. It is detected in GCs of secondary and antral follicles in adult life. Mitochondria holds an essential role in determining the developmental competence of oocytes. The BCL-2 family mediates the intrinsic pathway, in which apoptosis plays a crucial role [25]. Upon the transmission of stress stimuli to the mitochondria, pro-apoptotic members of the BCL-2 family, *BAX* and *BAK*, enhance the mitochondrial membrane’s permeability to protein like cytochrome c. This, in turn, triggers the activation of the caspase cascade [26].

Two genes related to apoptosis regulation, namely, *BCL-2* and *BAX*, showed lower expression levels, but their ratio significantly increased in a cohort study conducted by Tsui et al. (2014); this finding implies that DHEA treatment could potentially augment the anti-apoptotic mechanisms in cumulus cells of patients with poor ovarian response [13]. Similar results were observed in the case-control study of our review by Lin et al. (2017). In women with poor ovarian response, DHEA supplementation led to a notable decrease in the mRNA levels of *BAX, BAD,* caspase-3, caspase-9, and cytochrome c in cumulus cells when compared to those not receiving DHEA treatment [16]. Additionally, *BCL-2* mRNA was higher in cumulus cells from women with a poor ovarian response who received DHEA supplementation. This was correlated with a reduced percentage of apoptotic cells when compared to the non-supplemented poor ovarian response group [16]. In that same study, the expression of the *TFAM* gene, along with mitochondrial dehydrogenase activity and mitocohondrial mass, were found the be elevated in cumulus cells from the poor ovarian response group that received DHEA supplementation. *TFAM* is a crucial protein that binds to mitochondrial DNA (mtDNA) and modulates mitochondrial transcription initiation. It also serves as a key regulator of mtDNA copy number. This result suggests that DHEA supplementation positively affects the mitochondrial function of CCs [16].

DHEA may be involved in the ageing process, given its steadily decreasing level with age. Cellular senescence, characterised by the irreversible suppression of cell proliferation in response to various stressors, has been identified as a potentially significant factor in the process of aging and the development of age-related diseases [27]. The SA-β-gal activity, characterized as β-gal activity detectable at pH 6.0 in senescent cells, is extensively employed as a senescence biomarker due to the simplicity of the assay technique and its apparent specificity for senescent cells [27,28]. Numerous trials that tested the anti-ageing effects of DHEA failed to show the DHEA supplementation effect on anti-ageing. However, one included case-control study in this review holds a different view. In the study conducted by Lin et al. (2017), the senescent phenotype of CCs exhibited improvement in older patients following DHEA supplementations. This group demonstrated a significantly lower percentage of SA-β-gal-positive cumulus cells in comparison to the older group that did not receive DHEA supplementation [18]. These findings indicate that DHEA could be a potential therapeutic agent for delaying ovarian aging, thereby enhancing oocyte yield and quality.

To date, several studies have assessed the efficacy of DHEA supplementation on clinical outcomes for women undergoing IVF and ICSI cycle. However, these studies have yielded inconsistent results. In a prospective cohort study conducted by Vlahos et al. (2015), no significant differences were observed in clinical pregnancy and live birth rates between groups that received DHEA supplementation prior to IVF and those that did not undergo pre-treatment with DHEA [29]. Nonetheless, the study revealed a statistically significant elevation in anti-Mullerian hormone levels and a reduction in baseline follicle stimulating hormone (*p* < 0.001 and *p* = 0.007, respectively) for the group that received DHEA supplementation for a minimum of 12 weeks [29]. Yeung et al. (2015) carried out a randomized controlled trial involving 72 sub-fertile women with expected normal ovarian response. The study demonstrated no significant differences in antral follicle count, ovarian response to a standard low dose of gonadotrophin stimulation, or the number of oocytes retrieved between the group that underwent 12 weeks of DHEA pre-treatment and the placebo group [30]. Wang et al. (2022) conducted a randomized controlled trial to evaluate the impact of DHEA pre-treatment before IVF on the live birth rates in 821 women with poor ovarian response. The study found no beneficial effect of DHEA compared to the placebo, as there were no significant differences in the number of retrieved oocytes, clinical pregnancy rates, pregnancy loss rates, and cumulative live birth rates between the two groups [31]. Similarly, in the only RCT included in this review, Narkwichean et al. (2017) revealed no significant difference in the clinical outcomes. Furthermore, the study indicated no enhancement in oocyte quality following DHEA treatment, as evidenced by a panel of 10-gene expression profiles. Molecular markers of oocyte quality showed no differences in cumulus cell and granulosa cell samples between the DHEA treatment and control groups [14].

However, there are other researchers who hold varying opinions regarding the impact of DHEA supplementation on clinical outcomes. Sonmezer et al. (2009) suggested that DHEA supplementation has the potential to improve ovarian response, decrease cycle cancellation rates, and enhance embryo quality in patients with poor ovarian response [32]. A similar result was observed in the prospective cohort study by Zangmo et al. (2014), where DHEA treatment resulted in higher numbers of retrieved oocytes, fertilised oocytes and overall number of embryos [33]. Artini et al. (2012) investigated the influence of DHEA on the follicular microenvironment and IVF outcomes in poor responder patients [34]. The findings revealed a statistically significant decrease in hypoxic inducible factor 1 levels in women who received DHEA treatment (14.76 ± 51.13 vs. 270.03 ± 262.18 pg/mL; *p* = 0.002). The DHEA group exhibited a significantly higher number of mature oocytes retrieved (0.50 ± 0.52 vs. 0.08 ± 0.29, *p* = 0.018) [34]. Wiser et al. observed a comparable outcome in a randomized trial involving 33 women with low ovarian reserve. DHEA supplementation (75 mg/day) was associated with a higher yield of oocytes and a significantly improved birth rate [35]. In a meta-analysis of eight trials conducted by Li et al. (2015), it was demonstrated that DHEA supplementation led to an increase in clinical pregnancy rates among women with diminished ovarian reserve, poor responders to IVF/ICSI, and women experiencing premature ovarian aging compared to those receiving a placebo or no treatment (RR: 2.13; 95% CI: 1.12–4.08) [36].

In our review, multiple included studies reported similar cycle outcomes, such as a significant increase in the number of oocytes retrieved, fertilization rate, and the number of day-3 embryos. However, all studies found no differences in stimulation parameters (duration of stimulation and total dose of gonadotropins used) and pregnancy outcomes (clinical pregnancy, ongoing live pregnancy, and live birth rates) [13,14,15,16,17,18,19].

## 5. Strength and Limitation

The importance of DHEA supplementation on various aspects oocyte quality, such as follicular growth and maturation, regulation of apoptosis, mitochondrial function, and molecular and metabolic mechanisms, has been extensively discussed. In addition, the clinical outcomes and doses of DHEA supplementation has been clearly summarized, providing an accessible and comprehensive overview of the current evidence. The review findings suggest that DHEA supplementation can improve oocyte quality, particularly for women of advanced age or with poor ovarian response, leading to better IVF outcomes. By enhancing various aspects of oocyte development and function, DHEA supplementation may offer a promising adjunct treatment to improve fertility in these specific patient populations.

However, in all the studies included in our review, only women who were categorised into the older age or POR group received DHEA supplementation before the IVF cycle. The only RCT in this review enrolled exclusively women with a predicted DOR. Most of the studies had a small sample size, resulting in low statistical power. In order to support the implementation of DHEA supplementation in standard practice for IVF patients, it is crucial to conduct large-scale, rigorously designed RCTs. However, achieving this goal is practically challenging, as most women seeking IVF treatment desire to conceive the soonest possible and may be reluctant to participate in a randomised trial with the potential of being assigned to the placebo group.

## 6. Gaps and Future Recommendation

As the current evidence is not yet conclusive, larger randomized controlled trials are needed to validate the efficacy of DHEA supplementation as an adjunct therapy in IVF protocols. These trials should be well-designed and have a large sample size to ensure the reliability of the results. There may also be other unidentified genes that play a significant role in the relationship between cumulus cells and oocytes, which can, in turn, affect IVF outcomes. This possibility underlines the importance of employing innovative research methods and patient recruitment strategies to address the existing knowledge gaps in this fields.

Future studies should consider incorporating the following elements in their study design to generate more robust evidence on DHEA supplementation in IVF treatment: (i) Stratification, to reduce potential bias and increase the statistical power of the study; (ii) Personalized treatment approaches, which could involve identifying key genetic markers that predict response to DHEA supplementation and tailoring treatment accordingly; (iii) Multi-centre collaboration, which can help increase the study population and improve the generalizability of the findings; (iv) Integration of omics technologies, where it could lead to a better understanding of the underlying molecular mechanisms and inform the development of more targeted and effective interventions.

By incorporating these strategies into future studies, researchers can address existing knowledge gaps and generate more robust evidence on the efficacy of DHEA supplementation as an adjunct treatment in IVF protocols, ultimately improving the quality of care and outcomes for patients undergoing fertility treatment.

## 7. Conclusions

In summary, our review, which includes one RCT, two prospective cohort studies, and four case-control studies, has identified the potential beneficial effect of DHEA supplementations on ovarian CCs in improving oocyte quality for women of advanced age or with poor ovarian response (as defined by the Bologna criteria). Its impact on normal ovarian responders (NOR) or younger women remains unknown. Although the current evidence suggests a positive impact of DHEA supplementation on various molecular and cellular markers related to oocyte quality, further well-designed, large-scale RCTS are necessary to validate these findings and conclusively establish the efficacy of DHEA as an adjunct treatment in IVF protocols for these specific patient population. It is possible that other unrevealed genes may be significant contributors to the interconnection between CCs and oocytes and result in different effects on IVF outcomes. This highlights the necessity for innovative approaches to study design and patient recruitment in order to address existing knowledge gaps and produce more robust evidence on the efficacy of DHEA supplementation in IVF treatment.

## Figures and Tables

**Figure 1 life-13-01237-f001:**
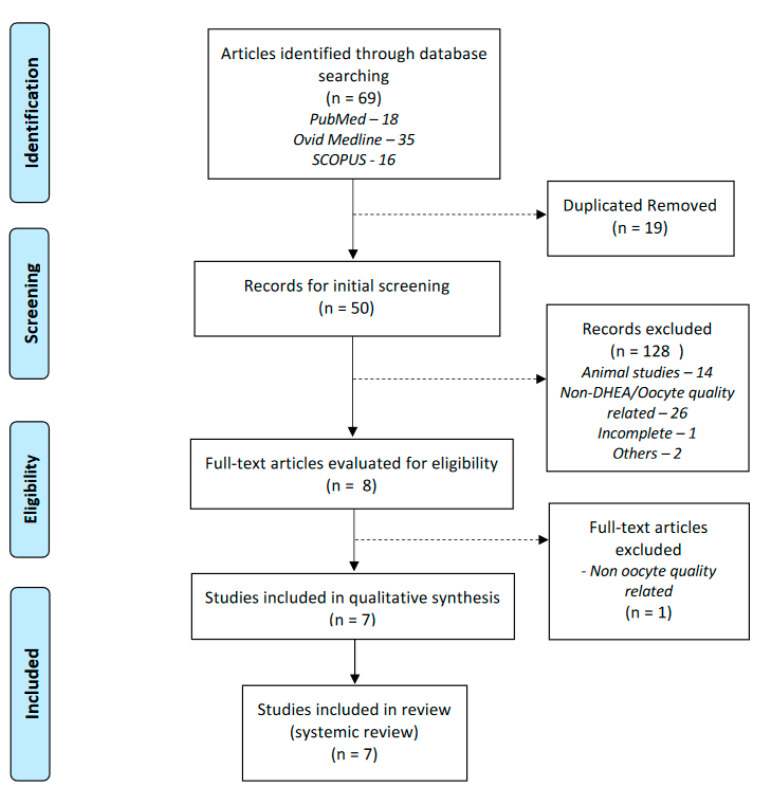
Prisma flow chart of the searching strategy.

## Data Availability

Data supporting this systematic review are available in the reference section. In addition, the analyzed data that were used during the current systematic review are available from the author on reasonable request.

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
