# Peer review of "Effects of Dehydroepiandrosterone (DHEA) Supplementation on Ovarian Cumulus Cells following In Vitro Fertilization (IVF)/Intra-Cytoplasmic Sperm Injection (ICSI) Treatment—A Systematic Review"

_life, 2023, doi:10.3390/life13061237_

Round 1

Author Response

Dear Editor,

Thank you for the reply with the reviewer’s report. We have taken into considerations of the comments provided by the reviewers and make the changes accordingly as per the following:

Reviewer’s Comment

Revision

Methodology: If details on risk of assessment can be added in results section of the paper, it will be helpful for readers

Table 4 to Table 6 added for Risk of bias assessment for all 3 diff study designs. (p13-15)

Results: Kindly add risk of bias assessment in the results

Table 4 to Table 6 added for Risk of bias assessment for all 3 diff study designs. (p13-15)

Discussion: Strength, future gaps and recommendations have to be highlighted

New section added: Strength & Limitation, Gaps and Future Recommendations. (p18 & p19)

Reviewer 2 Report

The study aims to explore the effectiveness of DHEA supplementation in ovarian cumulus cells following IVF/ICSI treatment.

I find the study difficult for me to understand it. I suggest that the auothors try to focus on few items and make it easier for the readers.

Author Response

Thank you for the reply. We have taken into considerations of the comments provided by the reviewers and make the changes accordingly as per the following:

Reviewer’s Comment

Revision

Methodology: If details on risk of assessment can be added in results section of the paper, it will be helpful for readers

Table 4 to Table 6 added for Risk of bias assessment for all 3 diff study designs. (p13-15)

Results: Kindly add risk of bias assessment in the results

Table 4 to Table 6 added for Risk of bias assessment for all 3 diff study designs. (p13-15)

Discussion: Strength, future gaps and recommendations have to be highlighted

New section added: Strength & Limitation, Gaps and Future Recommendations. (p18 & p19)

Reviewer 3 Report

Dear Authors,

I read your work and from my point of view is a well documented research. It respects the general criteria of a systematic review and it is appropriate written. The scientific value is considerable.

Author Response

(The authors gave the same response as above.)

Reviewer 4 Report

The manuscript titled "Effects of Dehydroepiandrosterone (DHEA) Supplementation on Ovarian Cumulus Cells following In-Vitro Fertilization (IVF)/ Intra-Cytoplasmic Sperm Injection (ICSI) treatment – A Systematic Review" has addressed the importance of DHEA supplementation for follicular health.

Although hyperandrogenism caused by DHEA, stimulates inflammation induced oxidative stress causing large follicular cyst in the ovary. In this manuscript, Woon SY et al. have well explained the role of hypoandrogenism in reduced ovarian function. They have broadly represented the importance of DHEA supplement on follicular growth and maturation, regulation of apoptosis, mitochondrial function- energy supply and molecular – metabolic mechanisms. In addition, clinical outcomes and supplementation of doses are clearly mentioned in the tables and the review concludes that DHEA supplementations improve the oocyte quality for women of advanced age or with poor ovarian response. Overall, the manuscript is well structured and informative. Hence, I strongly support this Systematic Review for publication.

However, I would suggest the authors to present the overview of the DHEA supplementation and its implications on ovarian function especially for follicular growth in the image format would made this manuscript a more reader friendly and easily understood their concept.

Author Response

(The authors gave the same response as above.)

Round 2

Reviewer 2 Report

The manuscript has improved and it might be published in the current form